

# A global data set of soil hydraulic properties and sub-grid variability of soil water retention and hydraulic conductivity curves

Carsten Montzka[1], Michael Herbst[1], Lutz Weihermüller[1], Anne Verhoef[2], Harry Vereecken[1]

[1]Forschungszentrum Jülich GmbH, Institute of Bio- and Geosciences: Agrosphere (IBG-3), Jülich, Germany
[2]University of Reading, Department of Geography and Environmental Science, Reading, UK

*Correspondence to*: Carsten Montzka (c.montzka@fz-juelich.de)

**Abstract.** Agroecosytem models, regional and global climate models, as well as numerical weather prediction models require adequate parameterization of soil hydraulic properties. These properties are fundamental for describing and

predicting water and energy exchange processes at the transition zone between solid Earth and Atmosphere, and regulate evapotranspiration, infiltration, and runoff generation. Hydraulic parameters describing the soil water retention (WRC) and hydraulic conductivity (HCC) curves are typically derived from soil texture via pedotransfer functions (PTFs). Resampling of those parameters for specific model grids is typically performed by different aggregation approaches such a spatial averaging and the use of dominant textural properties or soil classes. These aggregation approaches introduce uncertainty,

bias and parameter inconsistencies throughout spatial scales due to nonlinear relationships between hydraulic parameters and soil texture. Therefore, we present a method to scale hydraulic parameters to individual model grids and provide a global data set that overcomes the mentioned problems. The approach is based on Miller-Miller scaling that fits the parameters of the WRC through all sub-grid WRCs to provide an effective parameterization for the grid cell at model resolution; at the same time it preserves the information of sub-grid variability of the water retention curve by deriving local scaling

parameters. Based on the Mualem van Genuchten approach we also derive the unsaturated hydraulic conductivity from the water retention functions, thereby assuming that the local parameters are also valid for this function. In addition, via the Miller-Miller scaling parameter $\lambda$, information on global sub-grid scaling variance is given that enables modellers to improve dynamical downscaling of (regional) climate models or to perturb hydraulic parameters for model ensemble output generation. The present analysis is based on the ROSETTA PTF of Schaap et al. (2001) applied to the SoilGrids1km data set

of Hengl et al. (2014). The example data set is provided at a global resolution of 0.25° at DOI:10.1594/PANGAEA.870605 (DOI registration in progress, so far the data can be accessed under https://doi.pangaea.de/10.1594/PANGAEA.870605).

**Keywords:** SoilGrids, pedotransfer function, aggregation, scaling, hydraulic properties, water retention, hydraulic conductivity





## 1 Introduction

Hydraulic properties have fundamental importance in the description of water, energy and carbon exchange processes between the land surface and the atmosphere (e.g. Ek and Cuenca (1994), Xue et al. (1996)). Therefore, agroecosystem models (e.g. SWAP; van Dam et al. (2008)) and land surface models (LSMs; see below) require adequate parameterization

of soil hydraulic properties, i.e., more specificly, the water retention curve (WRC) and the hydraulic conductivity curve (HCC). These properties regulate the relative magnitude of water balance fluxes such as evapotranspiration, infiltration and surface and sub-surface runoff (Vereecken et al., 2016), and thus the amount of water held in the soil at any one time. As well as affecting the water balance components, they also play a role in the land surface energy balance; directly via their effect on latent heat flux (evapotranspiration, e.g. via available soil water influencing plant water stress, see Verhoef and

Egea (2014)), and indirectly because the size of the evapotranspiration co-determines land surface temperature, that in turn affects net radiation, sensible heat flux and soil heat flux (the latter is affected also via soil moisture dependency of soil thermal properties). With regards to the carbon balance, photosynthesis and soil respiration both strongly depend on soil moisture content and hence implicitly on choice of soil hydraulic models and their parameters.

State-of-the-art LSMs, e.g., NOAH (Niu et al., 2011), CLM (Oleson et al., 2008), VIC (Liang et al., 1994), JULES (Best et

al., 2011) and ORCHIDEE (Ngo-Duc et al., 2007) are key components of regional and global climate models (RCMs/GCMs) and numerical weather prediction models (NWPMs). They form important components of reanalyses (e.g. ERA-Interim/Land; Balsamo et al. (2015)) and model-data assimilation systems, such as the NASA Land Information System (LIS) or the Global Land Data Assimilation System (GLDAS; Rodell et al. (2004)). LSMs solve Richard´s equation for the water flow in the saturated/unsaturated zone. A fundamental problem is the adequate parameterization of the water

retention and hydraulic conductivity function to solve Richard´s equation. At point scale, a broad suite of experimental methods are available that allow measuring the WRC and HCC. These measurements are however expensive and time consuming, and often comprise intensive field sampling campaigns. Alternatively, parameters of the WRC and HCC can be estimated from in-situ or remotely sensed data in combination with parameter estimation techniques (Scharnagl et al., 2011;Bauer et al., 2012;Dimitrov et al., 2014;Jadoon et al., 2012;Montzka et al., 2011). At larger scales, such as those where

RCMS and GCMs are employed, the WRC and HCC are impossible to estimate (because underlying soil properties vary widely within grid cells) and are unobtainable by direct measurements.

To overcome this problem, the estimation of the required soil hydraulic properties is usually based on pedotransfer functions (PTFs) that use simple soil properties such as texture, organic matter, and bulk density to derive the parameters of mathematical equations that describe the HCC and the WRC (Vereecken et al., 2010). The idea behind PTFs is that more

easily available soil data such as soil texture, soil organic carbon content, or bulk density can be used to predict the hydraulic parameters for the WRC and HCC. In the last three decades, soil scientists have developed a broad suite of PTFs that differ with respect to the parameterizations of soil hydraulic properties for which they are used, the type of soil properties needed as inputs to derive the model-dependent parameters, and their spatial patterns. PTFs were developed for the prediction of



parameters used in the Campbell (1974) family of hydraulic functions (e.g., Clapp and Hornberger (1978)), Brooks and Corey (1964) (e.g., Rawls and Brakensiek (1985)), and Mualem van Genuchten equations (e.g., Rawls and Brakensiek (1985); Vereecken et al. (1989); Scheinost et al. (1997); Wösten et al. (1999); Weynants et al. (2009); Toth et al. (2015)). Bouma (1989) distinguished between two types of PTFs, namely continuous and class type PTFs. Continuous PTFs use

information on textural properties, bulk density, and soil organic matter amongst others, whereas class type PTFs do not estimate the parameters based on continuous textural and other soil properties but estimate the parameters for defined textural classes (e.g., 12 USDA textural classes) (Clapp and Hornberger, 1978;Toth et al., 2013). The disadvantage of the latter is that only the class average can be predicted and inner-class variability is neglected.

Another issue with the soil hydraulic parametrization in LSMs is caused by the spatial resolution of the model application

under consideration (e.g. GCM model runs for reanalyses or NWP model runs for weather forecasts), which is currently several (tens of) kilometres. This means that the soil input parameters to be provided at this scale have to be derived from existing data sources. Unfortunately, intrinsic soil properties are highly variable in space; in most cases several soil types can be found within a single grid cell of GCMs, for example. These soil types often differ strongly in soil texture, soil organic carbon content, and bulk density, as well as in soil depth and layering. Consequently, the fine scale soil information,

available from state-of-the-art soil maps such as the European LUCAS (Land Use/Land Cover Area Frame Survey) (Toth et al., 2013;Ballabio et al., 2016) at 500m resolution or the global SoilGrid database at 1km resolution (Hengl et al., 2014), have to be up-scaled to the scale at which the LSMs are being employed. The general problem of up-scaling, or change in spatial resolution of the input data by aggregating small scale input data, and the resulting output uncertainty for various model states was reported e.g., by Cale et al. (1983), Rastetter et al. (1992), Pierce and Running (1995), Hoffmann et al.

(2016), and Kuhnert et al. (2016). A practical example is GLDAS2-NOAH, where the porosity and the percentages of sand, silt, and clay at the original scale of the input data from Reynolds et al. (2000) were horizontally resampled, i.e. spatially averaged, to the 0.25° GLDAS grid (Rodell et al., 2004). Despite their importance, only a few studies investigating the implications of the above-mentioned issues (up-scaling, aggregation or resampling) on the model results have been conducted in the past.

The most straight forward method to aggregate input parameters from small-scale soil maps to larger scale grid cells of the GCMs would be spatial averaging. For some soil properties such as soil organic carbon, bulk density, or soil depth this kind of approach seems reasonable, whereas for soil texture averaging it is associated with considerable problems. For example, a GCMs grid cell containing  a pure sand soil for half of its area with the other half a pure clay would provide a sandy clay on average, which neither adequately reflects the sand nor the clay soil physical properties. Additionally, averaging percentages

may cause artefacts in closing the mass balance. A second method would be the averaging of soil hydraulic parameters (e.g., the van Genuchten parameters), whereby Zhu and Mohanty (2002) clearly showed that averaging of especially the shape parameters (*a* and *n*) is associated with considerable uncertainty. A third and most widely used aggregation technique for soil inputs at coarse model resolution is the one based on dominant soil types, where the dominant soil type within a coarse grid cell is derived from the fine scale soil map. However, in using this approach some information will get lost in the GCM



model outputs because non-dominant, but physically very differently behaving soils will not be taken into account during the model runs. In consequence, fluxes from non-dominant areas of the grid-cell will not be reproduced at large scale.

The theory introduced by Miller and Miller (1956) provided a technique to scale the relationships of pressure head and hydraulic conductivity by considering microscopic laws for capillary pressure forces and viscous flow in porous media based on a similarity assumption of the pore space structure (Warrick et al., 1977). Similarity scaling allows converting hydraulic characteristics (e.g. pressure head or conductivity) of one system (e.g. a soil sample) or location (e.g. a point scale measurement of WRC at field scale) towards corresponding characteristics of another system or location (Tillotson and Nielsen, 1984) under the condition that  the internal geometry of the system only differs by a characterize size. Miller-Miller scaling therefore allows capturing the spatial variability of soil hydraulic properties in one single scaling parameter rather than having to specify the statistics for each single hydraulic parameter (Warrick et al., 1977). The set of scaling factors (i.e. each location or sample has one scaling factor) follow approximately a log-normal distribution (Simmons et al., 1979). Tuli et al. (2001) analysed a physically based scaling approach of unsaturated hydraulic conductivity and soil water retention functions from pore-size distribution. They assumed that the relationship between both characteristics is log-normally distributed and that pores are geometrically similar, and showed that in this case scaling factors computed from median pore size or capillary pressure head can be used to describe the variability of unsaturated hydraulic conductivity functions. Using a fractal model, Pachepsky et al. (1995) showed that the spatial variability of water retention functions could be described by the spatial variability of a single dimension-less parameter. Ahuja et al. (1984) found that scaling factors for different soil depths are also related, and Clausnitzer et al. (1992) investigated the potential to simultaneously scale the WRC and HCC and found evidence that the results do not necessarily require independent scaling. In more detail, Hendrayanto et al. (2000) showed that separate scaling resulted in large estimation errors in either effective saturation or hydraulic conductivity. Further scaling methods have been developed based on the fractal method, e.g. the piece-wise fractal approach proposed by Millan and Gonzalez-Posada (2005) or the wavelet transform modulus maxima introduced by Zeleke and Si (2007). Wang et al. (2009), as well as Fallico et al. (2010), analysed the multifractal distribution of scaling parameters for soil water retention characteristics. Shu et al. (2008) stressed the need for location-dependent scale analyses to improve the performance for soil water retention characteristic predictions. Jana and Mohanty (2011) showed that a Bayesian Neural Network can be applied across spatial scales to approximate fine-scale soil hydraulic properties. With this approach ground-, air-, and space-based remotely sensed geophysical parameters directly contribute to a PTF in a single processing step instead of aggregating/scaling the estimated parameters to other scales in an independent second step. Recently, Fang et al. (2016) established an amplification factor for soil hydraulic conductivity to compensate for the resulting retardation of water flow due to the loss of information content as consequence of spatial aggregation. Liao et al. (2014) pointed out that uncertainty in the soil water retention parameters mainly results from the limited number of samples used for deriving PTFs and the spatial interpolation of basic soil properties. However, the latter error contribution dominates the potential to correctly determine spatial parameters, which leads to the assumption for our study that existing PTFs provide adequate parameters for global model applications. Nevertheless, the scaling uncertainty still needs to be considered.

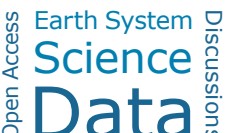

The objectives of this study are therefore: (i) to apply the Miller-Miller scaling approach to the state-of-the-art soil data set SoilGrids1km to provide a global consistent soil hydraulic parameterisation for GCMs based on first principles. These soil hydraulic parameterizations can be used in models of the terrestrial system to predict soil water fluxes based on solving

Richard´s equation from local to global scale, (ii) to present a method to identify the sub-grid variability of WRC and HCC with reference to the 1 km resolution SoilGrids soil texture data base. This scaling information can be used to perturb hydraulic parameters to generate ensemble runs with GCMs or to improve GCM downscaling, (iii) to evaluate the performance of the scaling approach for the calculation of the WRCs and HCCs against standard aggregation procedures based on two exemplary grid cells with varying variability in textural properties, and finally (iv) to demonstrate the

importance of the scaling variance at different spatial resolutions for three larger regions in North America, Africa, and Asia. We provide the corresponding aggregated global data set for the ROSETTA PTF (Schaap et al., 2001) at 0.25° regular grid spacing at DOI:10.1594/PANGAEA.870605 (DOI registration in progress, so far the data can be accessed under https://doi.pangaea.de/10.1594/PANGAEA.870605).

## 2 Material and Methods

In this section the data base and the approach to generate the global data set of soil hydraulic parameters is presented (see also Figure 1). In the following, all analyses are based on the SoilGrids 1km data base in combination with ROSETTA (Schaap et al., 2001). Other soil data bases and PTFs can be used similarly. The PTF will be applied to the high resolution soil texture maps to predict high resolution Mualem-van Genuchten (MvG) parameters. These are then scaled to calculate

soil hydraulic properties for the WRCs and HCCs employed at a coarser scale. In Section 2.3, the scaling approach is explained, and additionally the treatment of the sub-grid variance will be presented. In section 2.4 details about the application to generate the global data set are given.

*Insert Figure 1 here*

### 2.1 SoilGrids 1km

The SoilGrids 1km data base (Hengl et al., 2014) is a consistent, coherent, and global data set created by automated mapping (Vereecken et al., 2016). The main inputs are publicly available soil profile data, such as the USA National Cooperative Soil Survey Soil Characterization database (NCSS), the Land Use/Cover Area frame Statistical Survey LUCAS (Toth et al.,

2013), and the Soil and Terrain Database (SOTER) (Van Engelen and Dijkshoorn, 2012). Moreover, additional information, derived from moderate-resolution imaging spectroradiometer (MODIS) satellite imagery and the Shuttle Radar Topography Mission (SRTM) digital elevation model, has been used. Artificial surfaces as well as bare rock areas, water bodies, shifting



sands, permanent snow and ice were neglected. The resulting soil properties at seven predefined depths (0, 5, 15, 30, 60, 100, and 200 cm) are: soil organic carbon (g kg$^{-1}$), soil pH, sand, silt, and clay fractions (%), coarse fragments (gravel) (%), bulk density (kg m$^{-3}$), cation-exchange capacity (cmol+ kg$^{-1}$), soil organic carbon stock (t ha$^{-1}$), depth to bedrock (cm), World Reference Base soil groups, and USDA Soil Taxonomy suborders. SoilGrids implements model-based geostatistics and

multiple linear regressions for predicting sand, silt, and clay percentages as well as bulk density; and General Linear Models with log-link function for predicting organic carbon content. Lower and upper confidence limits at 90% probability of the predictions are also provided. In theory, the full prediction uncertainty could be used to estimate soil hydraulic property uncertainty but in our study we restricted the analysis on the mean predicted values. The findings of Hengl et al. indicate, that the distribution of soil organic carbon content in horizontal direction is mainly controlled by climatic conditions

(temperatures and precipitation), while the distribution of texture is mainly controlled by topography and lithology. The advantage of SoilGrids over other soil data-bases is that it provides pixel-based information rather than gridded vector information from class-based vector polygons. It should be noted that SoilGrids is stored in a World Geodetic System 84 (WGS84) regular grid with 1km resolution at the equator. The resolution at other latitudes is therefore higher.

**2.2 Pedotransfer functions, water retention and hydraulic conductivity functions**

A global high resolution hydraulic parameters data set is needed to infer the scaling characteristics for large scale climate models. Based on the textural information stored in the SoilGrids data base, PTFs can directly estimate the required hydraulic parameters (Figure 1). Several PTFs have been developed; here, we focus on the widely used PTF ROSETTA by Schaap et al. (2001), which is based on Neural Network predictions for the estimation of the Mualem van Genuchten (MvG)

parameters $\theta_s$, $\theta_r$, $\alpha$, $n$, $K_s$, and $L$ (van Genuchten, 1980), whereby the WRC to describe the effective volumetric saturation $S_e$ is calculated according to:

$$S_e(h) = \frac{\theta - \theta_r}{\theta_s - \theta_r} \begin{cases} 1 & h \geq 0 \\ [1 + (\alpha|h|)^n]^{-m} & h < 0, \ \alpha, m > 0, \ n > 1 \end{cases} , \qquad (1)$$

where $\theta_r$ [cm$^3$ cm$^{-3}$], and $\theta_s$ [cm$^3$ cm$^{-3}$] are the residual and saturated volumetric water content, respectively, and $\alpha$ [cm$^{-1}$], $n$ [-] and $m$ [-] ($m = n - \frac{1}{n}$) are shape parameters. Finally, the MvG approach to describe the HCC is given by:

$$K_r(h) = K_s S_e^L \left[ 1 - \left( 1 - S_e^{1/m} \right)^m \right]^2 \qquad (2)$$





where $K_r$ [-] is the relative hydraulic conductivity and $K_s$ [cm d⁻¹] is the saturated hydraulic conductivity, and $L$ is the pore connectivity parameter [-].

In a first step, ROSETTA was used to predict the MvG parameters based on the textural information of the SoilGrid map for each 1km cell. In a next step, the water retention pairs ($S_e$ versus $h$) were calculated for predefined pressure heads $h$ [cm] using the pressure head vector $\boldsymbol{h}$:

$$\boldsymbol{h} = [-1, -5, -10, -20, -30, -40, -50, -60, -70, -90, -110, -130, -150, -170, -210,$$
$$-300, -345, -690, -1020, -5100, -15300, -20000, -100000, -1000000] \tag{3}$$

The pressure heads in $\boldsymbol{h}$ were chosen to reflect pressure steps commonly used in laboratory analysis. We assumed -300 cm to reflect field capacity whereas wilting point ($h\sim$ -1500 cm) is generally found between -1020 and -5100. In pF terms ($\log_{10}(\boldsymbol{h})$) the $\boldsymbol{h}$ vector went up to 6.

### 2.3 Scaling approach and sub-grid variability estimation

In this study, the Miller and Miller (1956) scaling approach is applied to the parameters derived from the SoilGrids data for each soil depth separately. The procedure characterizes scaling factors to relate the hydraulic properties at a specific location to the mean hydraulic properties at a reference point or a point representative for a larger region.

In a first step we need to find adequate parameters for the retention function at the coarse scale (Figure 1). This approach has been reported in Clausnitzer et al. (1992). For each subpixel $i$ the relative saturation $S_{e_i}$ is calculated by:

$$S_{e_i}(\boldsymbol{h}) = f(\boldsymbol{h}, \alpha_i, n_i) = [1 + (\alpha_i|\boldsymbol{h}|)^{n_i}]^{-m_i} \tag{4}$$

Next, the coarse-scale parameters $\hat{\alpha}$ and $\hat{n}$ of the water retention curve $f(h, \alpha_i, n_i)$ need to be found that minimize the sum of squares of the deviations for all respective subpixels $i = 1 \dots N$, with $N$ being the number of subpixels within the coarse grid
cell:

$$(\hat{\alpha}, \hat{n}) = \text{argmin}_{\alpha,n} \sum_{i=1}^{N} [S_{e_i} - f(\boldsymbol{h}, \alpha_i, n_i)]^2 \tag{5}$$

The parameter fitting algorithm used in this study was the damped least-squares method of Levenberg–Marquardt (LM)
(Marquardt, 1963) to find a global minimum. As initial values for LM fitting the grid-specific spatial average of $\alpha$ ($\bar{\alpha}$) and $n$ ($\bar{n}$) was used.





Russo and Bresler (1980), as well as Warrick et al. (1977), showed that scale factors for soil water retention and unsaturated hydraulic conductivity are not necessarily identical. However, Clausnitzer et al. (1992) reported that a independent fitting would lead to inconsistencies in the parameter space, and that a single scaling factor is well-suited to describe the distribution and correlation structure of HCC and WRC. In this study we use the relationship between $K_r$ and $S_e$ in Eq. (2). Therefore, the scaling of the WRC can be directly transferred to the HCC, by using $\hat{\alpha}, \hat{n}$ from Eq. 5, even though these scaling parameters might not be the optimum choice for the HCC. However, this approach was chosen for simplicity; to allow for easy handling of hydraulic parameters via a single scaling parameter for global Earth system model applications. For the coarse grid cell representative HCC the missing parameters $K_s$ and $L$ are spatially averaged from the sub-grid parameters, in case of $K_s$ in logarithmic space. Similarly, also $\theta_s$ and $\theta_r$ were spatially averaged, i.e. for the coarse resolution $\overline{\theta_s}$ and $\overline{\theta_r}$ were calculated.

In a second step, the sub-grid variability is estimated by introducing the scale parameter $\lambda$ to the hydraulic head vector to simplify the description of the statistical variation of soil properties (Figure 1). This is done by

$$\boldsymbol{h^*} = \frac{\boldsymbol{h}}{\lambda}. \tag{6}$$

After substituting $\boldsymbol{h}$ by $\boldsymbol{h^*}$ in the van Genuchten (1980) water retention function (Eq. 4), while using previously estimated $\hat{\alpha}$ and $\hat{n}$, only the scaling factor is fitted for each individual subpixel. Eq. 5 can then be rewritten as:

$$\left(\widehat{\lambda_i}\right) = \text{argmin}_{\lambda_i} \sum_{i=1}^{N} \left[S_{e_i} - f(\mathbf{h}, \hat{\alpha}, \hat{n}, \lambda_i)\right]^2 \tag{7}$$

Equation 7 is subject to the constraint that the coarse grid mean of the set of scaling factors is unity $\left(\frac{1}{N}\sum_{i=1}^{N} \log_{10}\left(\widehat{\lambda_i}\right) = 0\right)$. This constraint is already approximated by adequately fitting of $\hat{\alpha}$ and $\hat{n}$ in the first step. Again, similar to the first step, the unsaturated HCC is scaled based on the parameters estimated for the WRC.

We recommend calculating the variance of the logarithmic $\widehat{\lambda_i}$ as a parameter of sub-grid variability for further use. The sub-grid variability information $\text{var}\left(\log_{10} \widehat{\lambda_i}\right)$ can be used in further research to perturb the soil hydraulic parameterization in ensemble runs of climate or weather prediction models.

## 2.4 Global application

The scaling method proposed here is applied to the parameters derived from the whole SoilGrids1km data set. In this study, every terrestrial coarse grid cell is identified with a unique ID, where the SoilGrids 1km attribution to the coarse cell was performed within a GIS system. This ensures flexibility to predict parameters for any type of grid, no matter whether it is



approximately isotropic, such as in the MetOffice Global Atmosphere 4.0 and Global Land 4.0 model (Walters et al., 2014), or an unstructured mesh of hexagonal/triangular grid cells in the Ocean-Land-Atmosphere Model (OLAM) (Walko and Avissar, 2008). We chose a spatial resolution of 0.25°, as it is used e.g. in GLDAS-NOAH (Rodell et al., 2004). One coarse grid cell contains exactly $30 \times 30 = 900$ fine resolution pixels. The number of fine resolution pixels used for calculating the scaling statistics is also provided with the data set, because lakes and broad rivers reduce the number of relevant pixels. In addition, for global application a land-sea mask has been established to omit irrelevant pixels. The final data set is delivered for latitudes ranging from -60° to 90°, omitting Antarctica.

### 2.5 Analysis procedure

In this section the procedure is explained concerning how the final data set of hydraulic parameters and scaling information was evaluated. This was done by selecting sample regions for a detailed presentation of the data set performance. Two coarse grid cells of different sub-grid heterogeneity were selected and the scaling results for the WRCs and HCCs compared discussed. The importance of considering sub-grid variability is stressed for different spatial resolutions by means of three larger regions.

### 2.5.1 Detailed grid cells analysis

In order to investigate the performance of the scaling approach in more detail, two coarse grid cell within Germany were selected based on an initial analysis of the sub-grid sand standard deviation (Figure 2).

*Insert Figure 2 here*

The focus on German sites is motivated by the large variation in soil texture, from a heterogeneous region in the North of Germany to a relatively homogeneous region in the German central lowlands. Moreover, the number of soil profile information contribution to the SoilGrids neural network approach is quite high in these regions. The first grid cell was selected in the South of Lower Saxony where Pleistocene morainal plains turn into Jurassic and Triassic rocks. This region exhibits small-scale differences in rocks and sediments where the soils developed from. The second region selected is located in the South East of North-Rhine Westfalia where the soil developed from Devonic weathered rocks as well as fluviatile sediments from the Rhine river system. This region can be regarded to be relatively homogeneous in soil texture. See also the soil texture diagrams in Figure 3.

*Insert Figure 3 here*





In order to compare our results to other commonly applied aggregation schemes, we also calculated the WRCs and HCCs (i) by averaging soil texture information and then using the ROSETTA equations; (ii) by averaging MvG parameters directly; ii) by identifying the dominant USDA soil class for each coarse grid cell and then utilizing class-representative MvG parameters; iii) by identifying the dominant USDA soil class for each coarse grid cell and then utilizing the Clapp and Hornberger (1978) approach to calculate the hydraulic properties, which requires dedicated hydraulic parameters. The Clapp and Hornberger (1978) parameterization is based on the Campbell approach (Campbell, 1974) for calculation of water retention and unsaturated hydraulic conductivity. This approach has been added to illustrate the differences between the VGM and Campbell (1974) approach that is still often used in GCM models. $S_{e\,Camp}$ after Campbell (1974) is given by:

$$S_{e\,Camp}(h) = \frac{\theta}{\theta_s} \begin{cases} 1 & h \geq \mathrm{h}_B \\ \left(\frac{1}{\mathrm{h}_B}|\boldsymbol{h}|\right)^{-\frac{1}{b}} & h < \mathrm{h}_B \end{cases} \qquad (8)$$

$\mathrm{h}_B$ is the air entry value, $b$ is the pore size distribution index [-]. The related hydraulic conductivity function for Campbell (1974) is given by:

$$K_{r,Camp}(h) = K_s \left(\frac{\theta(h)}{\theta_s}\right)^{(3+2b)} \qquad (9)$$

Note that the Campbell approach is similar to the Brooks and Corey (1964) approach, the only difference is that the latter did not set $\theta_r = 0$ in the WRC. Also Clapp and Hornberger (1978) did not consider $\theta_r$ in their equations.

**2.5.2 Analysis of scaling variability for different spatial resolutions**

In order to stress the importance of considering the sub-grid scaling information provided by the proposed method, the mean $\mathrm{var}\left(\log_{10} \hat{\lambda}_i\right)$ is quantified for different spatial resolutions. Three regions were analyzed in more detail, which are regions in North America, central Africa, and China/Mongolia consisting of 2048×2048 fine pixel from the SoilGrids 1km database (see Figure 4). For each of these larger regions a single parameter set of $\hat{\alpha}$ and $\hat{n}$ is estimated by using Eq. 5. For each fine 1 km pixel the scaling parameter $\hat{\lambda}_i$ is calculated according to Eq. 7. Different resolutions of 2, 4, 8, 16, 32, 64, 128, 256, 512, and 1024 km were applied to the resulting map of $\hat{\lambda}_i$. For each spatial resolution the cell-specific scaling parameter is averaged $(\mathrm{mean}\left(\log_{10} \hat{\lambda}_i\right))$. Finally, the variance of these averaged scaling parameters is calculated for the large regions. Herein we hypothesise that the variance of the scaling parameter is a function of spatial resolution.

*Insert Figure 4 here*



This analysis uses the original SoilGrids1km spatial resolution as a reference. However, it has to be mentioned that the calculated scaling variability only refers to the information content of the SoilGrid 1 km reference, and does not necessarily provide the real soil scaling variability.

## 3 Results and discussion

For brevity in this paper the results for the top soil layer are discussed only. After a description of the global data set, this section discusses the results for the two example pixels in Germany and the influence of different spatial resolutions on the variability of three larger regions.

### 3.1 Global analysis

The resulting global hydraulic parameters $\hat{\alpha}$, $\hat{n}$, $\bar{\theta}_s$ as well as $\text{mean}(\log_{10} K_s)$ are presented in Figure 5. Parameter $\hat{\alpha}$ ranges between 0.0036 and 0.045 cm$^{-1}$ with a global average of 0.0143 cm$^{-1}$ and shows a clear biogeographical or climatic related distribution. Relatively low values can be found mainly under boreal forests, but also in the North China Plain, Central

Europe, US Midwest, and the Cordoba province in Argentina. Especially the low bulk density of boreal top soils in the SoilGrids data base, and their widespread occurence, reduce the global average of $\hat{\alpha}$ to that low value. Relatively high values of $\hat{\alpha}$ were found at locations with high sand fractions, such as the desert regions of Sahara, Namib, the Arabian peninsula, and to a lesser extent also in Australia. Smaller regions with high $\hat{\alpha}$ could be traced in Florida and the morainal plains of Northern Europe and the Rocky Mountains. The global average of $\hat{n}$ is 1.547, with a range between 1.174 and 4.33 [-]. The

extreme high $\hat{n}$ values are found only in the non-alluvial regions of Sahara and Rub' al Khali ('Empty Quarter', Arab peninsula). The relatively high global average of $\hat{n}$ is caused by the same effect that caused the low $\hat{\alpha}$ average in the low bulk density of boreal top soils. Those soils typically are characterised by high organic carbon contents, behave quite differently in hydraulic sense compared to more mineral dominated soils, and are rarely used to develop classical PTFs. Therefore, independent from the aggregation or up-scaling approaches, more research is needed to adequately parameterise boreal soils

by appropriate PTFs.

*Insert Figure 5 here*

The global map of mean saturated hydraulic conductivity ($\text{mean}(\log_{10} K_s)$) in Figure 5 ranges between 0.174 and 3.105

with a global average of 1.784 (cm d$^{-1}$). Low soil saturated hydraulic conductivities are located in India, the Sahel, the Mediterranean, Central Asia, Levant and Iran, Texas, the US prairie regions, California, and South Central Canada. Highest



mean($\log_{10} K_s$) are found in Sahara and Rub' al Khali, where sandy soils dominate, but also in the upper Amazon basin (Marthews et al., 2014) and in cold climates.

*Insert Figure 6 here*

Figure 6 shows the global map of sub-grid scaling variance calculated from SoilGrids1km data set with a 0.25° grid reference. Global var$\left(\log_{10} \hat{\lambda}_i\right)$ ranges between ~0 and 1.574, with an average value of 0.229. This shows that large regions, e.g. Siberia, East China, Southern coastal provinces of Brazil, and Mexico, are relatively homogeneous within individual grid cells. The regions with particularly high sub-grid variability are typically transition zones of soils evolved from young

10  sediments. Examples are the holocene morainal plains in Northern Europe and Canada, as well as the slopes of high mountains areas of the Andes and Himalayas. Similarly, young deposits of large rivers such as the Amazon, and the inner Congo basin are characterized by high variability. These are the regions where the consideration of the sub-grid variability may have strong impacts on weather prediction and climate simulations.

The final data set is stored in netcdf format in WGS84 projection and contains the information given in Table 1.

Table 1: Variables stored in the final data set. The variable **z** indicates the soil depth, i.e., $z \in [0, 5, 15, 30, 60, 100, 200]$ cm.

| Variable | Units | Explanation | Variable name |
|---|---|---|---|
| Latitude | Decimal degree | Latitude in degrees North, Southern hemisphere in negative numbers | latitude |
| Longitude | Decimal degree | Longitude in degrees East, West of Greenwich in negative numbers | longitude |
| $\hat{\alpha}$ | cm$^{-1}$ | Fitted $\alpha$ at **z** cm depth for MvG parameterization | alpha_fit_**z**cm |
| $\hat{n}$ | - | Fitted $n$ at **z** cm depth for MvG parameterization | n_fit_**z**cm |
| $\overline{\theta_s}$ | m$^3$m$^{-3}$ | Mean $\theta_s$ at **z** cm depth for MvG parameterization | mean_theta_s_**z**cm |
| $\overline{\theta_r}$ | m$^3$m$^{-3}$ | Mean $\theta_r$ at **z** cm depth for MvG parameterization | mean_theta_r_**z**cm |
| $\overline{L}$ | - | Mean pore connectivity parameter at **z** cm depth for MvG parameterization | mean_L_**z**cm |
| $(\mathbf{mean(\log_{10} K_s)})$ | cm d$^{-1}$ | Mean saturated hydraulic conductivity at **z**cm depth | mean_Ks_**z**cm |
| $\mathbf{var\left(\log_{10} \hat{\lambda}_i\right)}$ | - | Scaling parameter variance at **z** cm depth | var_scaling_**z**cm |
| Valid subpixels | - | Number of valid subpixels for calculating scaling statistics for the **z** cm soil depth | valid_subpixels_**z**cm |

### 3.2 Example pixels

The detailed analysis of the WRC of two example model grid cells after applying a range of typical aggregation methods is shown in Figure 7. This includes aggregation by averaging soil texture information, averaging MvG soil hydraulic parameters, and by selecting the dominant USDA soil class (both with MvG and Campbell equations). Averaging soil

texture and averaging MvG parameters to a coarser grid caused differences in the WRC for Lower Saxony but showed nearly the same curve for North-Rhine Westfalia. The reasons why textural and MvG parameter averaging yielded the same aggregated WRC in North-Rhine Westfalia are unclear but as Zhu and Mohanty (2002) pointed out the reliability of MvG averaging greatly depends on the fine-scale (here 1km) heterogeneity (textural differences), but also on the textural class. Therefore, in the heterogeneous region of Lower Saxony the difference is large which leads to 0.04 higher $S_e$ at 2.5 $\log_{10}(h)$

when averaging texture as opposed to averaging MvG parameters. However, more drastic differences occur when using the dominant USDA soil class to predict coarse scale WRCs, where the Clapp and Hornberger parameterization for the Campbell model results in wetter conditions for all pressure heads than when using the class-representative MvG parameters and the MvG model.

*Insert Figure 7 here*

Finally, the WRCs based on the different aggregation approaches (dominant soil class, texture averaging, Miller-Miller scaling) are presented in Figure 8 for the two example regions. As can be seen, the effective saturation ($S_{e_i}$) calculated for the 1km sub-pixels at the standard heads of the head vector $\boldsymbol{h}$ (Eq. 3) (blue dots) reflect the natural variability in soil texture

and corresponding soil hydraulic properties of the two example regions. This subscale variability is also captured by the standard deviation of the Miller-Miller scaling approach, which is larger for Lower Saxony as compared to North-Rhine Westfalia. This indicates nicely that Miller-Miller scaling is an appropriate approach to capture fine scale variability and to propagate the fine-scale uncertainty into the larger scale of interest.

*Insert Figure 8 here*

A similar pattern of the different scaling methods can be also found for the HCCs presented in Figure 9. By this Miller-Miller scaling approach, large scale modelling can capture not only the right aggregated 'mean' WRCs and HCCs but also the uncertainty can be taken into account by running the models for the retention/hydraulic characteristics spanned by the

variance of the scaling factor.

*Insert Figure 9 here*

Finally, we plotted the histograms of the calculated sub-grid scaling parameters (Figure 10). Here it has to be noted that the scaling parameter λ was log-transformed with zero mean. Again, the differences in sub-grid heterogeneity of the two example regions become obvious with lager variability in λ for Lower Saxony and lower variability for North-Rhine Westfalia.

*Insert Figure 10 here*

### 3.3 Analysing scaling variability for different spatial resolutions

For the analysis of the scaling variability for different spatial resolutions the mean variance ($\text{var}(\log_{10} \widehat{\lambda}_l)$) was calculated for different grid resolutions and plotted in Figure 11. For all three case sites (North America, Africa, and China, see also Fig 3) $\text{var}(\log_{10} \widehat{\lambda}_l)$ decreased with decreasing spatial resolution; the absolute scaling parameter variability of the Africa region is largest. This can be explained by the fact that the Sahel is prone to strong seasonal dry-wet cycles, which induces large variability in soil development (Da Costa et al., 2015). On the other hand, the relative decrease of variance with coarser
resolution for this region is compared to the other ones (Figure 11 right panel).

*Insert Figure 11 here*

Interestingly, ~90 % of the variance is still maintained at 16 km grid size, ~80% at 64 km, and ~70 % at 128 km. Even at 256
km spatial resolution, more than 50 % of the variability is accounted for, but a diverging trend between the regions is detectable.

### 4 Conclusions and outlook

Reliable soil hydraulic parameterization is important for global climate model predictions, including climate reanalyses, and
weather forecast models. State-of-the-art global soil maps can provide basic soil properties at sub-kilometre scale resolution. The transfer of these data towards coarser resolution hydraulic properties has been a topic in soil/land surface up-scaling or aggregation research for several decades. In this paper we present a scaling method based on Miller-Miller similarity theory which was applied to the SoilGrids1km data set to provide parameters for the van Genuchten model of water retention curve (WRC) and the Mualem-van Genuchten model of hydraulic conductivity curve (HCC). These curves are required to solve
variably saturated flow in soils using Richards´ equation. In addition, the sub-grid variability of both WRC and HCC is assessed, which can be of use for model ensemble generation in climate and weather forecast models, or for down-/up-scaling approaches. The final global data set at 0.25° spatial resolution is available at DOI:10.1594/PANGAEA.870605.



The new data set is presented and analysed at the global scale and in more detail by two different individual pixels differing strongly in textural composition. In comparison to aggregation by using dominant USDA soil classes, averaging soil texture and averaging soil hydraulic parameters, the curve fitting approach provides better estimates of coarse scale water retention and conductivity curves and related parameters. Moreover, the Miller-Miller scaling provides an indicator of sub-grid

variability, which is not available from the other methods mentioned above. For three regions different spatial resolutions were analysed according to their ability to represent the soil hydraulic variability of the original SoilGrids data base at 1km resolution. For all regions a common general loss of variability was observed, with loosing ~10 % of the variance at 16 km grid size, ~20% at 64 km, and ~30 % at 128 km.

The presented analysis has been conducted on two-dimensional soil maps, without consideration of vertical relationships

between soil layers or horizons. This approach can be easily extended towards a 3D scaling that honors the vertical spatial dependency. A follow-up paper will assess the impact of this data set on water and energy fluxes at the soil surface for global simulations. Similarly, the effect of using other PTFs than Schaap et al. (2001) needs to be evaluated on the global scale as well as the uncertainties introduced during pedotransfer on the scaling parameterization. We plan to provide similar data sets for other PTFs, e.g. of Rawls and Brakensiek (1985), Wösten et al. (1999), Weynants et al. (2009), and Vereecken

et al. (1989). A similar approach is planned to provide parameters for the Brooks and Corey equation.

**5 Acknowledgements**

This study was supported by the Helmholtz-Alliance on "Remote Sensing and Earth System Dynamics". The SoilGrids data set can be accessed at www.soilgrids.org.

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







**Figure 1: Proposed method to aggregate soil hydraulic properties and sub-grid variability of soil water retention and hydraulic conductivity curves.**





**Figure 2: Location of the 0.25° test pixels in Germany, and their sand fraction based on SoilGrids1km.**



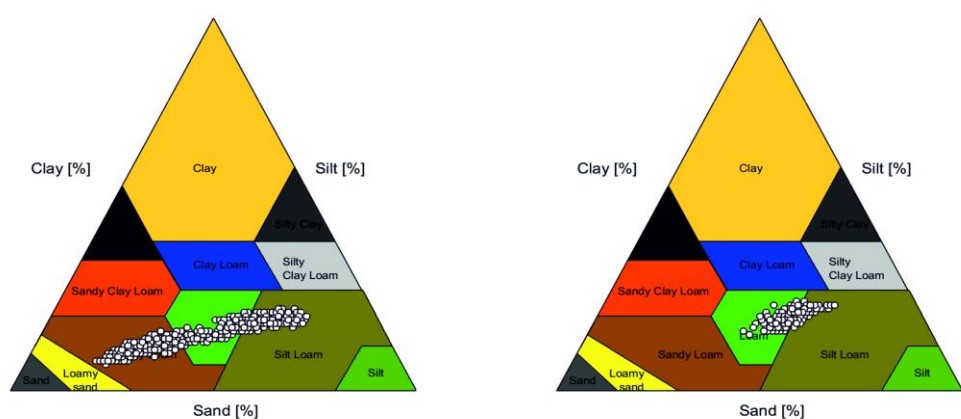

**Figure 3: Soil texture triangles, illustrating the difference in soil textural variability of the Lower Saxony pixel (left) and the North-Rhine Westfalia pixel (right), according to USDA classification.**

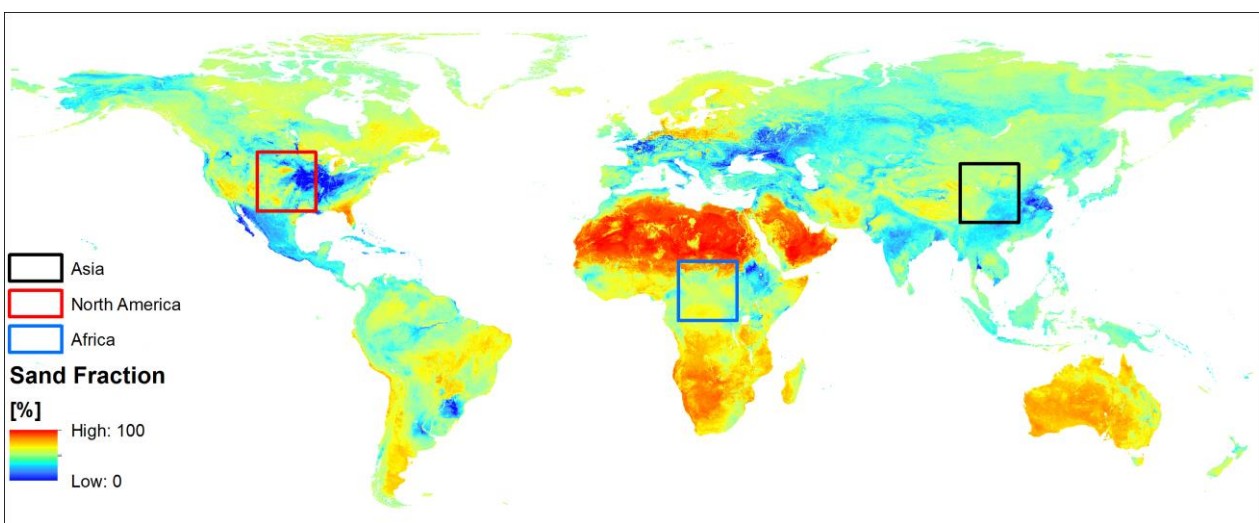

**Figure 4: Example regions selected to evaluate the scaling variance loss from Miller-Miller scaling at different spatial resolutions when neglecting the scaling variance. The background shows the sand fraction from SoilGrids1km.**





**Figure 5: Global map of $\widehat{\alpha}$ (top), $\widehat{n}$ (top middle), $\overline{\theta_s}$ (bottom middle) and mean($\log_{10}(Ks)$) (bottom) as derived when using the SoilGrids1km data set as input to the Rosetta PTF (Schaap et al., 2001), at 0.25° spatial resolution.**



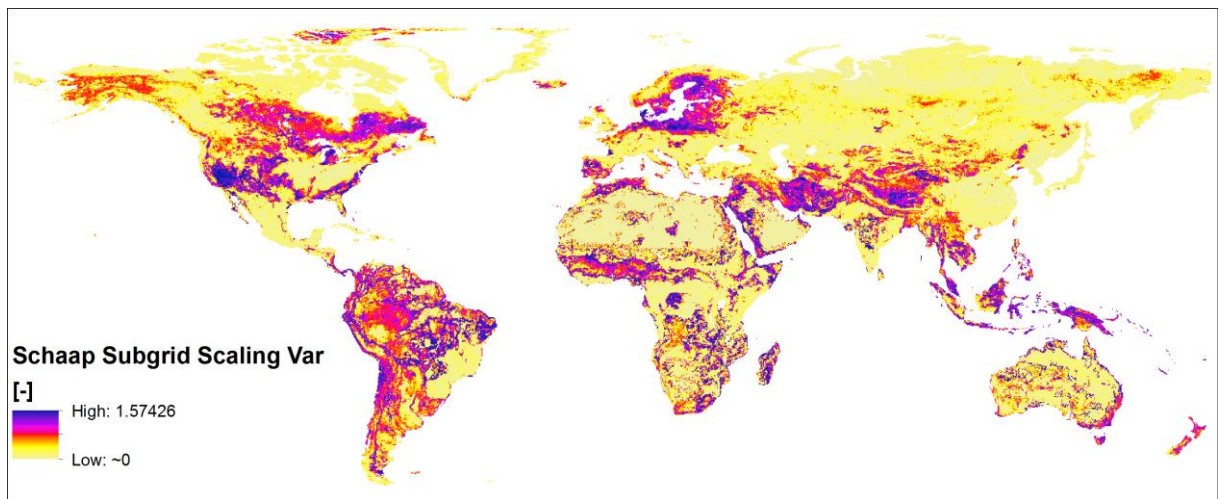

**Figure 6: Global map of** $\mathrm{var}\left(\log_{10}\widehat{\lambda}_t\right)$ **calculated from SoilGrids1km data set and the Rosetta PTF (Schaap et al., 2001) for 0.25°**
**resolution.**

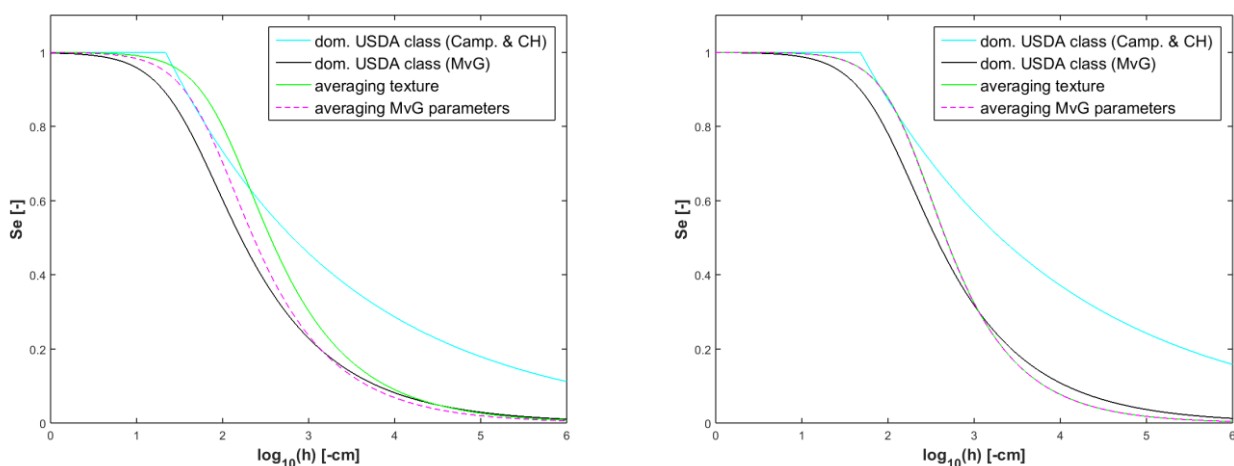

**Figure 7: PF curve after applying typical scaling or aggregation methods: by dominant USDA soil class and Clapp & Hornberger**
**parameters for the Brooks & Corey equation, by dominant USDA soil class and MvG equation, by averaging soil texture**
**parameters and then applying ROSETTA PTFs, and by averaging MvG soil hydraulic parameters directly, for Lower Saxony**
**(left) and North Rhine-Westfalia (right).**



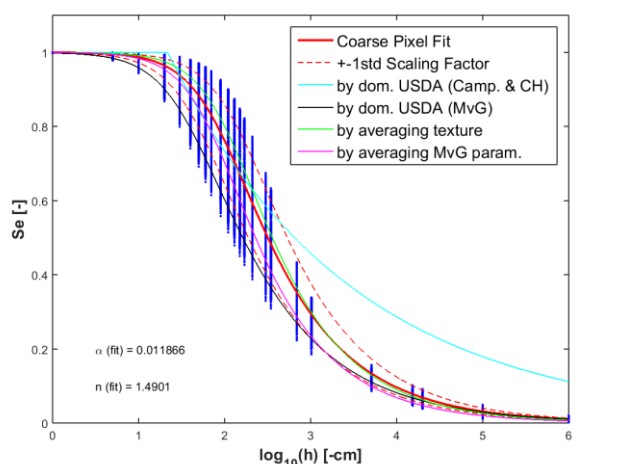
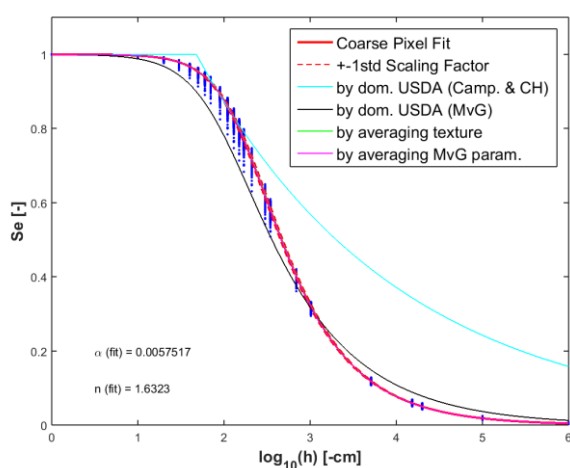

**Figure 8: Retention curves for Lower Saxony (left) and North-Rhine Westfalia (right) calculated using different approaches. The coarse pixel fit is the result of the Miller-Miller scaling approach, where also $\left(\log_{10} \hat{\lambda}_t\right)$ was estimated. Therefore, also $\pm 1\text{std}\left(\log_{10} \hat{\lambda}_t\right)$ could be provided to identify the sub-grid uncertainty. Further retention curves were calculated by dominant USDA class (both for Brooks & Corey with Clapp & Hornberger and for MvG equation), by averaging texture and by averaging MvG parameters. Blue points indicate $S_e$ at standard hydraulic heads for each individual subpixel.**

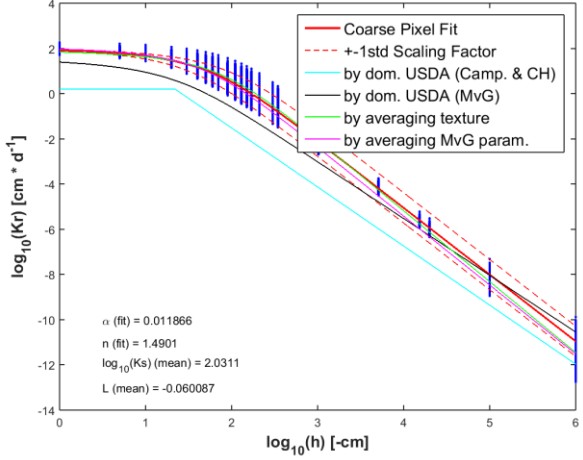
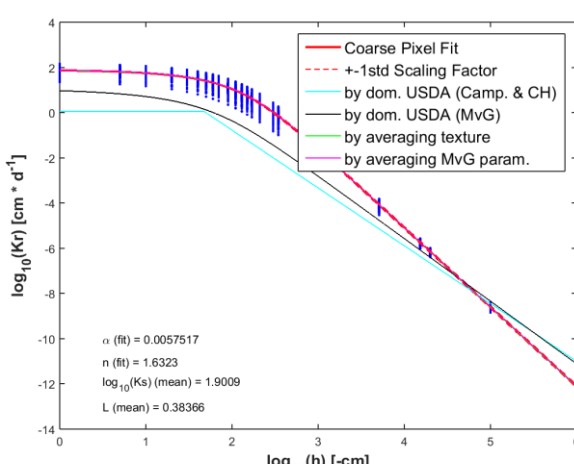

**Figure 9: Hydraulic conductivity curves for Lower Saxony (left) and North-Rhine Westfalia (right). The coarse pixel fit is the result of the Miller-Miller scaling approach, where also $\left(\log_{10} \hat{\lambda}_t\right)$ was estimated. Therefore, also $\pm 1\text{std}\left(\log_{10} \hat{\lambda}_t\right)$ could be provided to identify the sub-grid uncertainty. Further hydraulic conductivity curves were calculated by dominant USDA class (both for Brooks & Corey with Clapp & Hornberger and for MvG equation), by averaging texture and by averaging MvG parameters. Blue points indicate $\log_{10}(K_r)$ at standard hydraulic heads for each individual subpixel.**



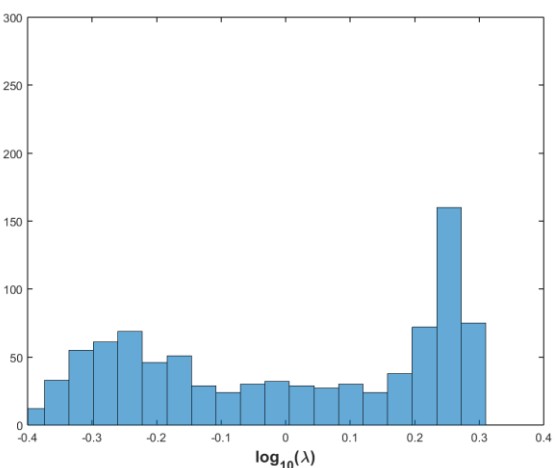 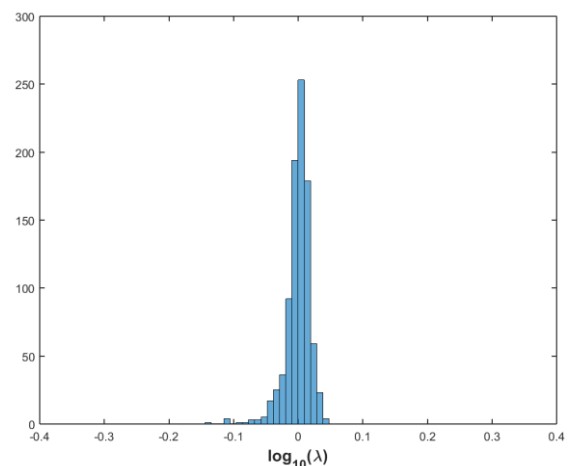

**Figure 10: Histograms of the retention scaling parameter ($\log_{10} \widehat{\lambda}_t$) for Lower Saxony (left) and North-Rhine Westfalia (right).**

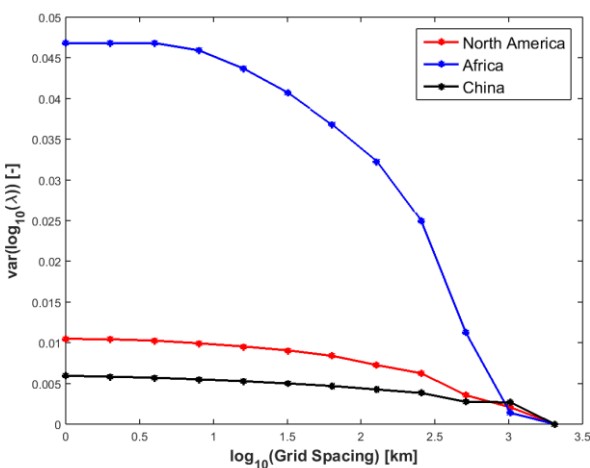 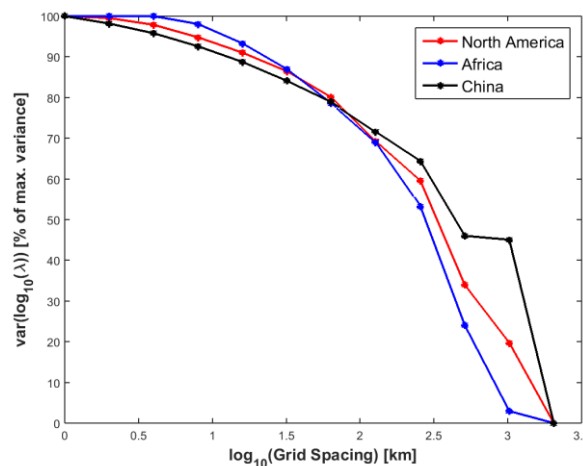

5 **Figure 11: Retention scaling parameter variance $\mathrm{var}(\log_{10} \widehat{\lambda}_t)$ for different grid resolutions and regions of interest (North America, Africa, and China, see also Figure 4). Left) absolute variance and right) variance normalised as percentage of the maximum variance at 1 km original SoilGrids resolution.**

