# Peer review of "A global data set of soil hydraulic properties and sub-grid variability of soil water retention and hydraulic conductivity curves"

_Earth System Science Data, 2017_

## Referee Comment (RC1) · Anonymous Referee #1 · 5 May 2017

Review of ESSD-2017-13, Montzka et al., Global Data Set of Soil Hydraulic Properties ….

Overall, a data set of very high quality and strong usefulness.  Very good fit to this journal.  Good descriptions and very good explanations.  Data set downloads easily from Pangaea.

Some additional explanation of the netCDF data files would help.  The 'Schaap' in the filename refers to application of the ROSETTA PTF (reference Schaap) but implies that one might in the future encounter a parallel data set produced using some other PTF strategy?  And the 'sl' refers to soil level but it took me a while to confirm that.  Some clearer naming or readme file could better help link these individual files to this description?

A few other comments and suggestions:

Many instances e.g page 2 line 24 where sequence of references lacks needed spaces between references (e.g Dimitrov et al., 2014;Jadoon et al., 2012;Montzka et al., 2011).  This comes from the bibliometric software used by the authors.  They should fix the problem rather than relying on proofreaders to catch all these small formatting errors?

On page 4, line 8, the use of the phrase 'character size' in this sentence opens some confusion?  Earlier in the sentence the authors wrote about hydraulic characteristics but here 'character' size refers to the magnitude of a soil property?  Or to grain size characteristics or pore size characteristics.  The reader doesn't understand what the authors intend here.

On page 6 lines 20 to 26 the MvG factors $\alpha$ and $\eta$ are introduced as scaling factors. But in fact both terms derive from measurable soil properties?  Fundamental physical parameters then applied as descriptive (e.g. scaling) factors?  One gets the impression here that soil scientists use them for mathematical convenience, not as measured properties.

Figure 1 provides a very useful graphical map of the data processing.  Likewise the manuscript itself provides very useful, well-documented descriptions.  But, as the authors know, and as careful reading exposes, the processes used here to assemble the global reference data set and then to support - statistically - up-scaling to GCM grid scales both rely on and introduce some additional uncertainty?  Having - and plotting in geographic space - the variance terms will as the authors say prove very valuable, but calculating variances does not remove or explain underlying uncertainties?  This comment does not criticise the work, only suggest that, perhaps in a short paragraph in the conclusions, the authors assemble some of the uncertainty terms they have mentioned throughout the manuscript, as useful reminder to (perhaps) less-familiar users about the need for some caution.  This cautionary paragraph could include the scaling uncertainty (page 4), the fact that the authors have (probably correctly) used mean values rather than full confidence interval data from the SoilGrids source (page 6), that they derive HCC from WRC rather than calculating independently (another necessary choice but a choice none-the-less, e.g page 8), their hypothesis / expectation that variance scales with spatial resolution (page 10), that they report here only data from top-most soil layers (page 11), that they recognise deficiencies in application of global PFT to specific boreal soils (again page 11), etc.  Again, the authors have presented and defended good choices necessary to make a very good product!  But, as part of their conclusions, they should remind readers / users of the uncertainties avoided or introduced by those choices. A short addition to the conclusion section?

Page 10, lines 3, 4 - some numbering problems in this section?  '(ii)' used twice?

Page 10, line 9 - 'VGM' a new term (if so need definition) or in fact a typo of MvG?

Page 14, lines 14, 15 - this sentence "On the other hand, the relative decrease of variance with coarser resolution for this region is compared to the other ones (Figure 11 right panel)."  The authors mean to write 'comparable' rather than "compared"?

Figures 2, 4 - Very useful plots of this 0.25 degree product on top of the original 1km SoilGrids product.  But the authors have chosen sand fraction and a very compressed colour scale in both figures.  Does sand fraction represent the obvious comparison parameter for both German and global sites?  Figure 3 indicates that most variance in soil properties for the German sites occurs along the sand-silt axis but from the compressed colour scales we can't tell whether the final product can distinguish 20%, 50%, 80%, etc.

Figures 5, 6 - This may represent a too-simple visual-only analysis, but I think the authors with their descriptive text for these two figures have allowed a question to arise.  Given the distinctly higher (relative to other large regions of the planet) variance (visual, not calculated!) in soil hydraulic properties in the region across north Africa (no other large region shows such consistent extreme values in 5 top through bottom), and the authors's plausible explanation of a strong wet-dry seasonal signal in the Sahel (page 14) and predominance of sandy soils in the Sahara (page 12), wouldn't we expect distinctly high values of scaling variance at least across the zonal boundaries of that north African region (e.g. Figure 11 left)?  It might be very useful to have the regional boxes from Figure 4 reproduced in Figure 6?  Perhaps north Africa represents a region of hydrologic extremes but of relative spatial homogeneity in soil properties?  Or a region of relative data sparsity?  Perhaps these figures will provoke similar questions from other readers?

Figure 7, 8 and 9.  This reader finds these plots very useful!  But, if averaging the MvG parameters represents the preferred approach, at least from this work, shouldn't the conclusions (top paragraph of page 15) and these figures make a strong case?  In the narrative and in the graphics?  A clearer graphical and textual message could better promote the strong outcomes of this work?

Figure 11 - here again the distinct question about Africa emerges.  Figure 11 left shows very high values for Africa but, to this reader's eye at least, Figure 6 does not support this?

Figure 11 right presents some very interesting challenges.  In the figure, and in the supporting text (page 14, lines 19 to 21), scaling variance remains quite large (the authors use the word 'conserved') out to 100 km?  A default resolution for earth system models used in CMIP6 might be 100 km.  Some modelling centres will run higher resolution at perhaps 25 km globally.  Ensemble work at global scales - which could and should take advantage of the variances recorded in this data set - will almost certainly occur with 100 km (or larger) resolution.  In those 25 km to 100 km spatial ranges, for these soil properties, the upscaled products preserve more than 70% of spatial variability information?  Steeper drops (reduced variance information fidelity) occur above 100 km?  This seems like a very useful conclusion which might deserve more attention in the manuscript?

---

## Referee Comment (RC2) · Anonymous Referee #2 · 27 May 2017

The problem of providing good soil parameters to LSMs is undoubtedly acute. The work is technically solid. Several conceptual aspects of the work have to be explicitly discussed. 1. The authors are estimating van Genuchten parameters. Does this mean that they are going to run the Richards equation for the spatial support scale of 40 x 40 km? Do they have any indications of the applicability of this model at this scale? If they do not then it would be prudent to state that and to outline the means of proving such applicability. 2. The authors base their estimates on the Rosetta PTF. They indicate that they plan to repeat this work for other PTFs such as Rawls-Brakensiek, HYPRES, etc. Why do they want to do that? How can a future user of their results decide which PTF should be used? The authors should either state that they will provide some

guidance on PTF selection or refrain from producing several of conceptually equal but numerically different datasets. 3. Although the authors claim that they use the MIller-Miller scaling, they do not use it. They use the Warrick's scaling. The difference is that M-M scaling is formulated in terms of water contents, and the Warrick's scaling is formulated in terms of saturation degree, and this is what this paper is using. 4. The Miller-Miller scaling so far was demonstrated for relatively small spatial extents. It was shown for these small extents that the scaling multiplier is lognormally distributed. However this never was shown for the large extents. Therefore the use of the log-transform for scaling multiplier has to be either justified or discussed. 5. The authors propose to average the saturated hydraulic conductivity using log transform, essentially using the geometric mean. The saturated hydraulic conductivity is known to scale with the extent. Neither arithmetic nor geometric mean follow such scaling. The authors should justify or at least discuss the selection of the geometric mean.

---

## Author Comment (AC1) · 19 Jun 2017

**Reply to comments of reviewer No. 1**

Comment:
Overall, a data set of very high quality and strong usefulness. Very good fit to this journal. Good descriptions and very good explanations. Data set downloads easily from Pangaea.
Reply:
We thank the reviewer for his very positive evaluation and his constructive comments.

Comment:
Some additional explanation of the netCDF data files would help. The 'Schaap' in the filename refers to application of the ROSETTA PTF (reference Schaap) but implies that one might in the future encounter a parallel data set produced using some other PTF strategy?
Reply:
We added the file naming convention to the manuscript. The reviewer is right that we plan to provide also further data sets for alternative PTFs, especially to extend the application for models using Brooks-Corey.

Comment:
And the 'sl' refers to soil level but it took me a while to confirm that. Some clearer naming or readme file could better help link these individual files to this description?
Reply:
We added the file naming convention to the manuscript.

Comment:
A few other comments and suggestions:
Many instances e.g page 2 line 24 where sequence of references lacks needed spaces between references (e.g Dimitrov et al., 2014;Jadoon et al., 2012;Montzka et al., 2011). This comes from the bibliometric software used by the authors. They should fix the problem rather than relying on proofreaders to catch all these small formatting errors?
Reply:
We are sorry for this inconvenience but this seems to be caused by the Endnote style file provided by the journal. We will check this with the journal.

Comment:
On page 4, line 8, the use of the phrase 'character size' in this sentence opens some confusion? Earlier in the sentence the authors wrote about hydraulic characteristics but here 'character' size refers to the magnitude of a soil property? Or to grain size characteristics or pore size characteristics. The reader doesn't understand what the authors intend here.
Reply:
We modified the sentence to: "...under the condition that the internal geometry of the system only differs by size"

Comment:
On page 6 lines 20 to 26 the MvG factors $\alpha$ and n are introduced as scaling factors. But in fact both terms derive from measurable soil properties? Fundamental physical parameters then applied as descriptive (e.g. scaling) factors? One gets the impression here that soil scientists use them for mathematical convenience, not as measured properties.
Reply:
The reviewer is right that both parameters can be estimated from measurements of the pairs of water content and pressure head. Therefore, they are generally assumed to be intrinsic soil physical parameters. Unfortunately, the second statement is not right. We did not use them only for mathematical convenience. What we have done (and it seems that the reviewer got the message right during the course of reading) is that the alpha value will be scaled by an additional factor introduced – the scaling factor lambda. As this is common theory in scaling we believe that this does not need any further discussion or explanation.

Additionally, the entire scaling concept has been extensively described in the manuscript and no additional queries have been raised by any of the reviewers.

Comment:
Figure 1 provides a very useful graphical map of the data processing. Likewise the manuscript itself provides very useful, well-documented descriptions.
Reply:
Thank you.

Comment:
But, as the authors know, and as careful reading exposes, the processes used here to assemble the global reference data set and then to support - statistically - up-scaling to GCM grid scales both rely on and introduce some additional uncertainty? Having - and plotting in geographic space - the variance terms will as the authors say prove very valuable, but calculating variances does not remove or explain underlying uncertainties? This comment does not criticise the work, only suggest that, perhaps in a short paragraph in the conclusions, the authors assemble some of the uncertainty terms they have mentioned throughout the manuscript, as useful reminder to (perhaps) less-familiar users about the need for some caution. This cautionary paragraph could include the scaling uncertainty (page 4), the fact that the authors have (probably correctly) used mean values rather than full confidence interval data from the SoilGrids source (page 6), that they derive HCC from WRC rather than calculating independently (another necessary choice but a choice none-the-less, e.g page 8), their hypothesis / expectation that variance scales with spatial resolution (page 10), that they report here only data from top-most soil layers (page 11), that they recognise deficiencies in application of global PFT to specific boreal soils (again page 11), etc. Again, the authors have presented and defended good choices necessary to make a very good product! But, as part of their conclusions, they should remind readers / users of the uncertainties avoided or introduced by those choices. A short addition to the conclusion section?
Reply:
In order to clarify all listed sources of uncertainty, we introduced a new section 3.4.

Comment:
Page 10, lines 3, 4 - some numbering problems in this section? '(ii)' used twice?
Reply:
Corrected.

Comment:
Page 10, line 9 - 'VGM' a new term (if so need definition) or in fact a typo of MvG?
Reply:
Corrected.

Comment:
Page 14, lines 14, 15 - this sentence "On the other hand, the relative decrease of variance with coarser resolution for this region is compared to the other ones (Figure 11 right panel)." The authors mean to write 'comparable' rather than "compared"?
Reply:
Indeed, corrected.

Comment:
Figures 2, 4 - Very useful plots of this 0.25 degree product on top of the original 1km SoilGrids product. But the authors have chosen sand fraction and a very compressed colour scale in both figures. Does sand fraction represent the obvious comparison parameter for both German and global sites? Figure 3 indicates that most variance in soil properties for the German sites occurs along the sand-silt axis but from the compressed colour scales we can't tell whether the final product can distinguish 20%, 50%, 80%, etc.

Reply:
We have chosen the sand fraction to exemplarily present the heterogeneity within the 0.25°
grid cell and also within the example regions to analyze the scaling variance loss. Therefore,
the variability is important, not the absolute sand fraction magnitude.

Comment:
Figures 5, 6 - This may represent a too-simple visual-only analysis, but I think the authors
with their descriptive text for these two figures have allowed a question to arise. Given the
distinctly higher (relative to other large regions of the planet) variance (visual, not calculated!)
in soil hydraulic properties in the region across north Africa (no other large region shows
such consistent extreme values in 5 top through bottom), and the authors's plausible
explanation of a strong wet-dry seasonal signal in the Sahel (page 14) and predominance of
sandy soils in the Sahara (page 12), wouldn't we expect distinctly high values of scaling
variance at least across the zonal boundaries of that north African region (e.g. Figure 11
left)? It might be very useful to have the regional boxes from Figure 4 reproduced in Figure
6? Perhaps north Africa represents a region of hydrologic extremes but of relative spatial
homogeneity in soil properties? Or a region of relative data sparsity? Perhaps these figures
will provoke similar questions from other readers?

Reply:
In the global maps of alpha n, etc. in Figure 5 the clear differentiation between Sahara and
Sahel is visible. If the spatial resolution of the target grid is very coarse (e.g. >500km), we of
course expect high scaling variance at the zonal boundaries from Sahara to Sahel. However,
here we calculate the scaling variance for a 0.25° grid where the transition zone with >500km
is of too large scale that the 0.25° grid can capture this phenomenon. In this case, more
smaller scale processes contribute to the sub-grid variability, e.g. related to river networks,
volcanic activity and heterogeneous glacial processes.

Comment:
Figure 7, 8 and 9. This reader finds these plots very useful! But, if averaging the MvG
parameters represents the preferred approach, at least from this work, shouldn't the
conclusions (top paragraph of page 15) and these figures make a strong case? In the
narrative and in the graphics? A clearer graphical and textual message could better promote
the strong outcomes of this work?

Reply:
We respectfully disagree that we stated that MvG parameter averaging should be the
preferred approach. In general, it is known that MvG parameters should NOT be averaged
(nor in the lab neither in the field). Therefore, we introduced the entire methodology to
average retention pairs and use the scaling factor to capture some of the uncertainty
associated. Nevertheless, we included MvG averaging as a reference (because MvG
averaging seems still a choice for some researchers not being deeply in soil physics). We
again checked the manuscript and did not found at any place that MvG averaging is the
preferable method.

Comment:
Figure 11 - here again the distinct question about Africa emerges. Figure 11 left shows very
high values for Africa but, to this reader's eye at least, Figure 6 does not support this?

Reply:
Figure 6 cannot support the high variability in Africa, because it was developed for presenting
the scaling variance at 0.25° grid only. The blue box in Figure 4 is the extent we analyze in
Figure 11, where we use different grid spacings to estimate a scaling relationship between
resolution and sub-grid variance. We clarified this in the manuscript.

Comment:
Figure 11 right presents some very interesting challenges. In the figure, and in the supporting
text (page 14, lines 19 to 21), scaling variance remains quite large (the authors use the word
'conserved') out to 100 km? A default resolution for earth system models used in CMIP6

might be 100 km. Some modelling centres will run higher resolution at perhaps 25 km globally. Ensemble work at global scales - which could and should take advantage of the variances recorded in this data set - will almost certainly occur with 100 km (or larger) resolution. In those 25 km to 100 km spatial ranges, for these soil properties, the upscaled products preserve more than 70% of spatial variability information? Steeper drops (reduced variance information fidelity) occur above 100 km? This seems like a very useful conclusion which might deserve more attention in the manuscript?

Reply:

We included a discussion of the impact of the results on the modeling community in the conclusions.

---

## Author Comment (AC2) · 19 Jun 2017

**Reply to comments of reviewer No. 2**

Comment:
The problem of providing good soil parameters to LSMs is undoubtedly acute. The work is technically solid.
Reply:
We thank the reviewer for their constructive comments.

Comment:
Several conceptual aspects of the work have to be explicitly discussed. 1. The authors are estimating van Genuchten parameters. Does this mean that they are going to run the Richards equation for the spatial support scale of 40 x 40 km? Do they have any indications of the applicability of this model at this scale? If they do not then it would be prudent to state that and to outline the means of proving such applicability.
Reply:
We fully agree with this statement that the scale discrepancy between horizontal resolution (e.g. tens of kilometers) and vertical resolution (e.g. centimeters or even millimeters) of many global applied models is an issue valued for extensive discussion. However, several globally applied land surface models, which are within the target audience of this data set, already apply Richards´ equation globally, or at scales larger than pedon-scale. Example models are NOAH-MP (Niu et al., 2011) and LEAF/OLAM (Walko et al., 2000). This issue has been discussed in detail before, e.g. by Beven (2001). Therefore, we do not discuss this in this manuscript.

References:
Niu, G.-Y., et al. (2011): The community Noah land surface model with multiparameterization options (Noah-MP): 1. Model description and evaluation with local-scale measurements. *J. Geophys. Res.*, 116, D12109
Walko, R. L., W. R. Cotton, G. Feingold, and B. Stevens (2000): Efficient computation of vapor and heat diffusion between hydrometeors in a numerical model.Atmos. Res., 53, 171–183.
Beven, K.: How far can we go in distributed hydrological modelling?, Hydrol. Earth Syst. Sci., 5, 1-12, 10.5194/hess-5-1-2001, 2001.

Comment:
2. The authors base their estimates on the Rosetta PTF. They indicate that they plan to repeat this work for other PTFs such as Rawls-Brakensiek, HYPRES, etc. Why do they want to do that? How can a future user of their results decide which PTF should be used? The authors should either state that they will provide some guidance on PTF selection or refrain from producing several of conceptually equal but numerically different datasets.
Reply:
A large number of land surface models exist that are all being used to infer global distributions of energy and water balance fluxes, for a wide range of applications. These models are run either in uncoupled mode (where they are simply driven by global fields of atmospheric variables) or are embedded in Global Circulation models that allow for feedback between the land surface and the atmosphere. All of these models (rightly or wrongly) use the Richard's equation at large scales to calculate water flow in soils and hence need hydraulic functions. These are either described using the Brooks and Corey or van Genuchten approach. These models therefore need hydraulic parameters, derived from PTFs derived for either Brooks and Corey or van Genuchten. Land surface models currently use Look-up tables or arbitrarily selected PTFs to derive these parameters. In our follow-up studies we aim to provide coherent, reliable and well documented parameter sets for both Brooks and Corey or van Genuchten that can be used by the modelers to test the impact of these selections on their model results, for example.

However, the application of further PTFs is future work and should not be discussed in this manuscript which is specifically focused on the application of the Rosetta PTF model H3 after Schaap et al., 2001. The valued decision of land surface modelers for a specific PTF is justified by their application, the focus region and model structure. Therefore, we plan to offer data sets for the main applied PTFs. With this data set we provide van Genuchten parameters, which cannot be used by models where the Brooks-Corey hydraulic model is implemented. Therefore, additional specific parameterization (e.g. based on Rawls) different from van Genuchten is still needed.

Comment:
3. Although the authors claim that they use the MIller-Miller scaling, they do not use it. They use the Warrick's scaling. The difference is that M-M scaling is formulated in terms of water contents, and the Warrick's scaling is formulated in terms of saturation degree, and this is what this paper is using.
Reply:
This is correct. The basic is Miller-Miller scaling under the assumption that porosity is constant. Warrick relaxed this assumption. We clarified this in the manuscript.

Comment:
4. The Miller-Miller scaling so far was demonstrated for relatively small spatial extents. It was shown for these small extents that the scaling multiplier is lognormally distributed. However this never was shown for the large extents. Therefore the use of the log-transform for scaling multiplier has to be either justified or discussed.
Reply:
We discuss this issue in the newly formed section 3.4 about inherent uncertainties.

Comment:
5. The authors propose to average the saturated hydraulic conductivity using log transform, essentially using the geometric mean. The saturated hydraulic conductivity is known to scale with the extent. Neither arithmetic nor geometric mean follow such scaling. The authors should justify or at least discuss the selection of the geometric mean.
Reply:
We discuss this issue in the newly formed section 3.4 about inherent uncertainties.